# Metalloporphyrin Metal–Organic Frameworks: Eminent Synthetic Strategies and Recent Practical Exploitations

**DOI:** 10.3390/molecules27154917

**Published:** 2022-08-02

**Authors:** Arash Ebrahimi, Lukáš Krivosudský

**Affiliations:** Department of Inorganic Chemistry, Faculty of Natural Sciences, Comenius University in Bratislava, Mlynská dolina, Ilkovičova 6, 842 15 Bratislava, Slovakia; ebrahimi4@uniba.sk

**Keywords:** metalloporphyrins, metal–organic frameworks, porphyrins, synthetic strategies, biomimetic, (photo-)catalysis, electrochemical utilization

## Abstract

The emergence of metal–organic frameworks (MOFs) in recent years has stimulated the interest of scientists working in this area as one of the most applicable archetypes of three-dimensional structures that can be used as promising materials in several applications including but not limited to (photo-)catalysis, sensing, separation, adsorption, biological and electrochemical efficiencies and so on. Not only do MOFs have their own specific versatile structures, tunable cavities, and remarkably high surface areas, but they also present many alternative procedures to overcome emerging obstacles. Since the discovery of such highly effective materials, they have been employed for multiple uses; additionally, the efforts towards the synthesis of MOFs with specific properties based on planned (template) synthesis have led to the construction of several promising types of MOFs possessing large biological or bioinspired ligands. Specifically, metalloporphyrin-based MOFs have been created where the porphyrin moieties are either incorporated as struts within the framework to form porphyrinic MOFs or encapsulated inside the cavities to construct porphyrin@MOFs which can combine the peerless properties of porphyrins and porous MOFs simultaneously. In this context, the main aim of this review was to highlight their structure, characteristics, and some of their prominent present-day applications.

## 1. Introduction

The recently emerged porous materials, metal–organic frameworks (MOFs)—typically formed from metal ions/clusters bridged by multidentate ligands in an extended framework—have provided solutions to tackle challenges in areas such as catalysis [1,2,3], gas storge/separation [4,5], biomimetic applications [6,7,8], drug delivery [9,10,11], electrochemical applications [12,13] and biomedical chemistry [14,15]. Moreover, the structures can be tuned by replacing or incorporating specific linkers or suitable unsaturated metal ions, in addition to adjusting the pore size and/or geometry which substantially influence their catalytic behaviors toward many substances participating in the reaction [16,17]. Amongst the main categories of MOFs, porphyrin-based MOFs have demonstrated themselves as a tangible material able to provide the properties of both MOFs and metalloporphyrin complexes in one scaffold [18]. Despite being synthesized less frequently than other types of MOFs and less often explored in research, they have had considerable impact on multiple fields, particularly biomimetic and biomedical ones as catalysts, owing to their resemblance to some molecules discovered in nature [19].

Tetrapyrrole ligands such as porphyrin (or porphine in the unsubstituted form) and related macrocycles chlorin and corrin are naturally occurring in several bioinorganic metal complexes. The ability of the planar or nearly planar tetradentate ring system to stabilize kinetically labile metal centers (i.e., Mg^II^, Ni^II^, Fe^II/III^, Co^II^) results in the selective formation of stable metal complexes containing an extensively conjugated π system (Figure 1).

Thus, metalloporphyrins, due to their abundance in nature, have been explored during the last decades. The synthetic bioinspired complexes resemble in structure, central atoms, and properties the most common naturally occurring biomolecules such as hemoglobin which transports oxygen in animal bodies, chlorophyll which acts as a light-scavenging antenna in photosynthesis inside plants, or vitamin B_12_ which is important for metabolism in the cells (Figure 2, Table 1) [20]. The combination of such favorable properties makes artificial metalloporphyrins highly suited for applications in photosynthesis [21], electrochemical [22], biosensing [23], biomedical [24] applications for tumor therapy [25] and bioimaging [26].

The crystal engineering of MOFs based on metalloporphyrins, or tetrapyrrole ligands in general, began in the early nineties [28]. The intercalation of tetrapyrrole ligands was recently recognized to be driven mostly by weak dispersive forces and either an offset or a proper π−π stacking with other components of the MOF [29,30,31,32,33]. Interestingly, there is usually a lack of a stronger specific interaction between the porphyrin sheets themselves. Therefore, the construction of a crystalline MOF depends very much on the additives and their ability to provide suitable interactions and binding with metalloporphyrin and/or other components [34,35,36,37,38].

On the other hand, further to metallization in their center [39], metalloporphyrin complexes can also be additionally peripherally functionalized at meso- or β-positions [40] (in Porphyrin, there are typically 12 positions that can be exchanged in the environs (Pyrrolic rings containing the eight β positions and the other four meso ones which are attributed to the methine substituents)) or even complexed by various (non-) transition metals providing versatile moieties which had led to their use as spacers to construct several porphyrin-based MOFs where they could serve either inside the pores (porphyrin@MOFs) [41] or as linker merged in the architecture throughout the framework (porphyrinic MOFs) [42]. The representative structures of synthetized derivatives of porphine are depicted in Figure 3. They benefit from the extended π conjugated system throughout the planar molecule, which moderates substitution of the metal ions and the functionalities of porphyrin itself [43]. Therefore, it illustrates exceptional electrochemical and photophysical exploitations while possessing an extraordinary chemical and physical durability. Additionally, porphyrins and their accessories normally have the strongest Soret band (400–450 nm) and a pack of steadily reduced Q-bands located somewhere in the range of 500 to 700 nm in the absorption spectrum [44]. These features have led them to be considered as one of the most significant organic chromophores with remarkable adsorption bands in the visible region.

Along this line of study, many of the literature reviews have chiefly concluded that one of the several ways to resolve these kinds of issues is the immobilization of porphyrin inside or by anchoring them to the surface and as struts in the MOF framework [45]. Such functionalization would result in a tremendous improvement in their stability (for central atoms without an ideal ionic radius (Table 1)) [27] and their catalytic performances when completely within the MOF architecture in comparison with their homogeneous counterparts [14]. Nonetheless, these kinds of inclusion not only would boost their stability and hamper their suicidal tendency for self-quenching, but also enhance the activation of some inert molecules [46] In this minireview, we described the pertinent synthetic processes by which porphyrin-based MOFs are fabricated, and some of their popular utilizations were succinctly discussed.

## 2. Synthesis Procedures of Porphyrin-Based MOFs

By either integrating porphyrins/metalloporphyrins inside pores freely in situ or by grafting on the surface using post-synthetic methods and/or as part of the network component, porphyrin-based MOFs could be easily constructed. However, downsizing MOFs to the nanoscale will even more profoundly develop their size-dependent properties when encapsulating or accompanying such proactive molecules for any related applications [14].

### 2.1. In Situ Method of Porph@MOFs Synthesis

Inspired by these facts, some promising routes to combine porphyrin derivatives into MOF frameworks emerged such as porphyrin@MOFs (porph@MOFs). These include the entrapment of (metallo-) porphyrins into the cavities or the decoration of the surface of MOFs where the former method can be performed using one-pot fabrication in situ, and the latter can be executed post-synthesis. In contrast to in situ assembly in which free porphyrin basis/metalloporphyrin are entrapped by MOF precursors (metal ion and ligand) by self-assembly simultaneously (ship-in-a-bottle) [46], the post-synthetic method which occurs by anchoring to the exterior or the inclusion of them inside the MOF pores is mainly based on weak chemical interactions such as hydrogen bonding, electrostatics, van der Waals forces and others, between the pre-obtained MOF and porphyrin base/metalloporphyrin [47]. The simplicity and straightforwardness of porphyrin entrapping by in situ formation led to this path being applied extensively by many researchers working in this field even though the post-functionalization requires the acknowledgement of some issues such as the creation of a suitable interaction between these two materials [14,20]. Parameters that should be considered before postsynthetic fabrication include the activation of MOF pores and channels by guest solvent removal inside the structure during synthesis to allow for the incorporation of porphyrin instead, the dimensions in terms of shape and size of the porphyrin encapsulated into the cavities should be appropriate, and the stimulus required to initiate bond construction between the porphyrin and MOF structure.

As described above, embedded porph@MOF can be prepared by the in situ mixing of pre-synthesized porphyrins and MOF reactants (metal ion salt and linkers). A series of metalloporphyrin-decorated Cu-based MOFs with a coral-like shape (named as M-TCPP@Cu) were obtained using a one-pot reaction strategy [48]. Accordingly, the resulting MOFs were developed through intermediate enrichment-enhanced conversion to assist the electrochemical reduction of CO_2_ to C_2_H_4_. The respective porph@MOFs were obtained by dissolving H_3_BTC and TCPP/M-TCPP in a mixture of ethanol and DMF followed by the addition of Cu(NO_3_)_2_·3H_2_O in aqueous solution in situ to produce M-TCPP@Cu-MOF (M = Fe, Co, Ni). An ionic Mn-metalloporphyrin was reported which is presented in Figure 4 that was encapsulated into the interior pores of ZIF-8 by a simple method through which all the precursors in DMF were sealed in a Teflon-lined autoclave and heated at 140 °C for 2 days [49]. The crystals were then used as heterogeneous catalysts for a cycloaddition reaction of CO_2_ with epoxides.

### 2.2. Postsynthetic Procedure of Porph@MOFs Fabrication

For the postsynthetic fabrication of functionalized porph@MOFs, they can be either physically absorbed on the exterior or captured into the cavities, and a specific MOF must be prepared in advance; subsequently, the previously formed porphyrins are incorporated to or grafted on the MOF structure. Next, the metal can be exchanged in porphyrin via controlled-immersion of the final material in a solution of metallic salts. A post-synthetic modification (PSM) of a porphyrin-engulfed MOF to enhance the selective adsorption of CO_2_ over CH_4_ was reported [50]. The trapped porphyrin used as a structure-directing agent to provide a “ship-in-a-bottle” mode led to template-based Cd-porph@MOM-11 (MOM; metal–organic materials). In Figure 5a, the effect of the exchange of some cationic organic guests such as H_2_ppz^2+^ with Li^+^ on the selectivity of the adsorption of H_2_ over N_2_ was assessed. Remarkably, the results presented that ppz (1,1′,4′,1″,4″,1″′-quaterphenyl-3,5,3″′,5″′-tetracarboxylate) demonstrated significant kinetic trap for both the N_2_ and H_2_ ads-des process whilst Li displayed an increment in the pore volume size and more importantly a relatively higher H_2_ isosteric adsorption heat [51]. In another case, noncatenated hydroxyl-substituted MOF were introduced and replaced by Li^+^ and Mg^2+^ ions to convert pendant alcohol to metal alkoxides in order to upgrade the H_2_ uptake reversibly (Figure 5b). Exchanging was performed via the immersion of as-fabricated MOF in THF solvent (Tetrahydrofuran) to replace the primarily occupied solvent DMF (N-N-Dimethylformamide). Afterwards, the stirring of the respective MOF in an excess of Li^+^[O(CH3)^3−^] in CH_3_ CN/THF solvents was performed to exchange Li^+^ ions which boosted the hydrogen adsorption ability of the MOF significantly [52]. The illustration in Figure 5c [53] indicates that the MOF of the formula Zn_2_(NDC)_2_(diPyNI) (NDC = 2,6-dicarboxylate, diPyNI = *N*,*N*′-di-(4-pyridyl)-1,4,5,8-naphthalenetetracarboxydiimide) was reduced by Li^0^. The interaction imposed by H_2_ − Li^+^ inside MOF pores improved its competence to adsorb H_2_ which was most likely increased by the augmented ligand polarizability and framework displacement. Furthermore, the experimental work in [50] with some modifications of a combination of the first two methods mentioned in Figure 5d submerged single crystals of the prefabricated Cd-porph@MOM-11 into metal chloride salt solutions, with meso-tetra(*N*-methyl-4-pyridyl) porphine tetratosylate (TMPyP) in methanol serving as a template for PSM, and formed a basis for MOF formation via single-crystal-to-single-crystal ion exchange processes.

### 2.3. Porphyrinic-Oriented MOFs

With regard to porphyrinic MOFs, porphyrin or metalloporphyrin functioning as an organic linker is one of the main components in the framework, which coordinates with secondary building units (SBUs). Accordingly, the choice of suitable shape, size and geometry that meet the desired pore structure to load substrate molecules and to catalyze several reactions efficiently on the surface depends entirely on the rational selection of porphyrins and SBUs. While the insertion of (metallo-) porphyrins not only equips MOFs with new functionality, they can also maintain or bring about far better stability and diversity across the building blocks. As a consequence, the catalytic performances of these types of porous coordination networks could be simply upgraded by regulating Lewis-acid metal active sites and designing well-qualified circumferential functionalities on metalloporphyrins [46]. In line with the previous statements, various parameters such as temperature, solvent, reaction time and the method as well as proper metal nodes and porphyrins chosen, could additionally determine the final product [14].

By applying peripherally functionalized porphyrin/metalloporphyrin as spacer or multidentate ligands directly with metal ions/clusters could lead to the formation of porphyrinic MOFs. Displayed clearly in Figure 6, the porphyrinic MOF PCN-222/MOF-545 (free base-H_4_TCPP and [Zr_6_(μ_3_-O)_8_(O)_8_]^8−^ node) was used to selectively oxidize 2-chloroethyl ethyl sulfide (CEES) to a less toxic 2-chloroethyl ethyl sulfoxide (CEESO) at room temperature and neutral pH. The photooxidation of this mustard-gas simulant under mild conditions by exploiting these porous materials as photosensitizer within a half-life of up to 13 min was found to be one of the most convenient methods for the detoxification of such a poisonous compound [54].

Facile insertion of H_3_PW_12_O_40_ inside the solvothermally prefabricated free-base PCN-222 MOF was investigated for the photocatalytic synthesis of some bioactive N-heterocycles such as Nifedipine, Nicardipine, Nicotinic acid (Vitamin B_3_), and Pyridoxine (Vitamin B_6_) under visible-LED light irradiation [55]. The porphyrinic Zr-MOF scaffold was constructed by employing TFA and BA as modifiers followed by post-modification with POM to construct the POM@PCN-222 composite.

## 3. Practical Applications of Metalloporphyrin Metal–Organic Frameworks

In fact, the use of metalloporphyrin MOFs for versatile applications mostly stems from (metallo-) porphyrins segments, as described earlier, as they have a square planar structure providing permanent π-electrons delocalization within porphyrin which leads to many potential applications regarding their various characteristics and wide-ranging functions. For example, solar cells, light harvesting [56] and molecular electronic originate from visible light absorption. The catalytic activities of metalloporphyrins indicates their suitability as (photo-) catalysts [16], electrocatalysts [42] and biomimetic catalysts [10]. Metal ions’ sensing and realization abilities can be achieved via the modulation of their optical and electronic nature derived from the coordination of the metal site and their axial connection to molecules. Last but not least, their similarity to several molecules operating in the core site of the vital proteins in humans, has led to them being frequently used for plenty of biological utilizations such as biocompatibility, imitating functions in numerous biological systems [9,23,46], effectual removal and longer resistance against tumors, as they have less side effects, and they have also been largely employed as photosensitizers for photodynamic therapy (PDT) [25,26]. Concomitantly, their fluorescence properties suggest that porphyrin-based photosensitizers are beneficial fluorescence imaging-guided therapy systems for tumor or a multitude of diseases [57,58]. Table 2 summarizes the most significant recent research concerned with the (photo-) catalysis, and electrochemical and biomedical applications of porh@MOFs constructions.

### 3.1. Efficacious Catalytic Utilization

Regarding catalytic traits, a series of highly stable mesoporous metalloporphyrin Fe-MOFs; PCN-600 [M-TCPP (M = Mn, Fe, Ni, Cu, Co)] was synthesized utilizing prefabricated [Fe_3_O(OOCCH_3_)_6_] as building blocks [79]. They also demonstrated high durability in aqueous solution with a pH in the region of 2–11 and exhibited extremely high stability even in basic media. Using PCN-600(Fe) as an efficient catalyst to mimic the peroxidase function in the co-oxidation of phenol and 4-AAP (4-Aminoantipyrine), it was found that they exhibit excellent activity in similar reactions. More recently, the photophysical characterization of remarkably water-persistent PCN-223 MOFs formed from free porphyrin bases, *meso*-tetrakis(4-carboxyphenyl) porphyrin (TCPP), has been studied by employing transient absorption spectroscopy to demonstrate its highly efficacious light harvesting and energy transfer ability throughout the framework [56]. Figure 7 vividly illustrates the acyl transfer reaction between pyridylcarbinol (PC) and *N*-acylimidazole (NAI) after employing isoreticular zirconium-based MOFs, which shows that the degree of catalysis, however, relies remarkably on both the identity of the PC and of the MOF. In fact, relative rates differ by as much as 20-fold [80]. For the first time, the C-H bond halogenation reactions of cyclohexane/cyclopentane [69] were performed successfully using the porphyrinic PCN-602 (Mn) structure in a basic system. The pyrazolate-based porphyrinic MOF rendered superior durability in various coordinating anions in basic media which are extensively utilized in several catalytic reactions. PCN-602(Mn^3+^) has been acknowledged to be an extremely efficacious heterogeneous C–H halogenation catalyst of inert hydrocarbons upon basic ambience [81].

### 3.2. Precious (Photo-)Electrocatalytic Exploitation

With respect to their photocatalytic and electrochemical properties, acid-base resistant Zr-phenolate metalloporphyrin scaffolds have been utilized for CO_2_ photoreduction under visible light irradiation [82]. Light harvesting uniqueness derives from porphyrinic units along with highly stable Zr-oxide chains; catalytically active metal ion centers also have significantly enhanced sorption and catalytic traits. Furthermore, amongst them, ZrPP-1-Co represented its catalytic competence in terms of entrapping CO_2_ into the pores effectually and also due to its high photocatalytic activity and selectivity over CH_4_ to reduce CO_2_ to CO practically upon visible light radiance [61].

Recently, the incorporation of the fullerene C_60_ into porph@MOFs found that, theoretically, it could increase photoelectric conductivity by preventing the delocalization of π-electrons donor−acceptor interactions which may reduce electric conductivity. TD-DFT calculations [83] revealed that the electron transfer from a porphyrin to C_60_ either through direct near-infrared transitions or via photoinduced electron-transfer under visible-light excitation not only substantially prolongs electron–hole recombination and charge separation lifetime, but it also enhances its optoelectronic properties considerably. In another survey, the successful fabrication of novel rare-earth metal USTC-8(In) porphyrin-based MOFs was demonstrated. The synthesized materials were investigated to determine their stability in harsh acidic-basic media and photocatalytic H_2_ production performances. Additionally, in this case the In(III) ions easily disassembled from the porphyrin rings by exerting light radiation and readily hampered the recombination of e–h (electron-hole recombination). Therefore, the photocatalytic proficiency of metalloporphyrin MOFs (with Cu, Ni, and Co) improved considerably [70]. As shown in Figure 8, the Co-TCPP and Mn-TCPP immobilized into the ZIF-67 framework and pyrolyzed in an argon atmosphere yielding a diverse series of Co@NC-x and Mn@NC-x MOFs, which can be used as electrocatalysts for oxidation–reduction reactions (ORR). The insertion of a metalloporphyrin in ZIF-67 makes much more of the Co(Mn) and N sources available and Co-Nx active sites accessible for electrocatalysis [59]. More recently [71], different metal substitutions (Fe, Co, Cu) of metalloporphyrin in the PCN-601 (Ni-TPP) framework were tested in a catalyzed CO_2_ photoreduction. The results indicated that PCN-601(FeTPP) and PCN-601(CoTPP) are ideal candidates for photocatalysis. It is noteworthy to express that the catalysis of PCN-601(FeTPP) would meaningfully outperform that of PCN-601(CoTPP).

### 3.3. Profitable Biomedical Manipulation

Concerning biological and biomimetic applications, recently, the loading of polyethylene glycol (PEG)-coated PCN-222 with a pro-oxidant drug, piperlongumine (PL), was effectively demonstrated to cure breast cancer cells by chem–photodynamic combination therapy. Interestingly, in this experiment, sonodynamic therapy methods were employed to generate reactive oxygen species (ROS) by executing a safe nanosonosensitizer MOF in order to heal the patients suffering from these kinds of fatal diseases which lead to exceptionally increased ROS production [68]. A vancomycin-incorporated PCN-224 with antibacterial properties of vancomycin and highly sensitive photodynamic therapy activity of PCN-224 as a dual antibacterial agent was used to combat Gram-positive bacteria such as S-aureus [72]. The results displayed synergistic antibacterial competence combined with specific targeted activities under white LED illumination, making it as a promising strategy for antimicrobial therapy.

In one of the most recent methods to overcome the restriction caused by oxidative damages to cellular components resulting from interruption in redox homeostasis, a synergistic strategy including chemodynamic therapy (CDT) combined with photodynamic therapy (PDT) generated by Fe (III)-TCPP and glutathione (GSH) under optical laser irradiation was performed in Xue’s lab which is represented in Figure 9 [65]. After being capped by silk fibroin (SF) on the surface to construct NMOF@SF, it was utilized to carry tirapazamine (TPZ) prodrug and deliver it to mediate the reduction of Fe(III) to Fe(II). Taking advantage of the high bioimmunity and treatment particularity of NMOF provided through the Fenton-like ability of Fe (II) and TCPP-moderating feature combination in MOF coupled with GSH alternation to GSSG (Glutathione disulfide), this procedure successfully contributed to completely eradicating tumor in vivo and in vitro.

## 4. Summary and Remarks

Principally, the remarkable tunability, considerable porosity, biocompatibility, high surface areas and biodegradability of MOFs render them as a novel applicable material in many areas of chemistry. At the same time, porphyrin-based metal–organic frameworks integration can modify the possible instability and self-oxidation (quenching) deriving from free porphyrins in physiological environments as well as improve the physiochemical traits by incorporating the peripheral functionalities or multiple metal ions either on porphyrins or MOFs in a single architecture. The porphyrin-based MOFs discussed here were classified as follows: (1) (metallo-) porph@MOFs where porphyrin and their metalized derivatives are encapsulated inside the pores of MOFs, (2) postsynthetic porphyrin-based MOFs in which porphyrin can either be grafted on the surface or entrapped within the pores and (3) porphyrinic MOFs constructed through the linking of (metallo-) porphyrin peripheral functionalities on α or β positions or by the insertion of the most commonly unsaturated transition metal ions coordinated chemically to the ligand to construct a 1, 2 or 3D network. These approaches not only improve their stability, but they also result in better performances in (photo-) catalysis, electrocatalysis and mimicking biological systems. With the above considerations in mind, the introduction of (metallo-) porphyrins to MOFs has inhibited self-destructive oxidation and more importantly fostered the stability and reactivity of porphyrin molecules when confronting harsh media efficiently.

## Figures and Tables

**Figure 1 molecules-27-04917-f001:**
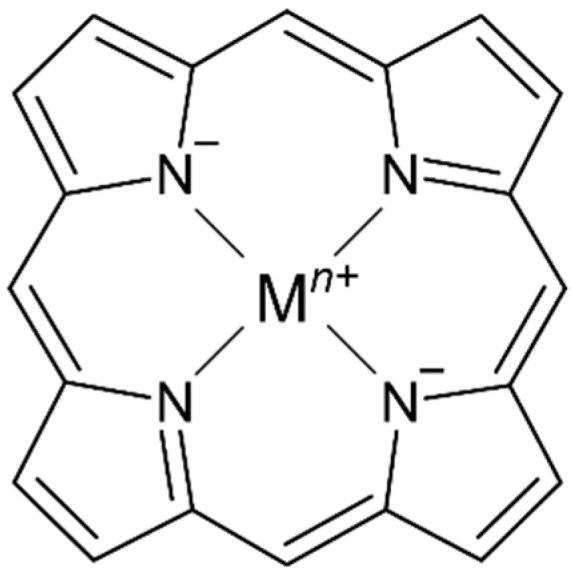
Representative structure of a metalloporphyrin complex with a porphine ligand core.

**Figure 2 molecules-27-04917-f002:**
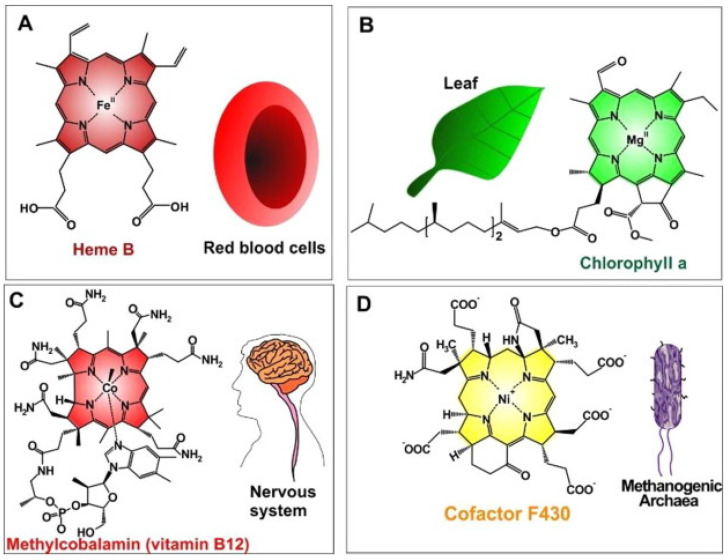
Naturally occurred MPs (metalloporphyrins) (**A**) iron(II)-porphyrin “Heme B in RBCs” to convey oxygen; (**B**) magnesium(II)-porphyrin “chlorophyll a” needed for plant photosynthesis; (**C**) cobalt(II)-porphyrin “methylcobalamin (as vitamin B12)” assisted to facilitate nerve system performances; (**D**) nickel(II)-porphyrin “Cofactor F430” accelerates methanogenesis in methanogenic archaea). Reprinted with permission from [20].

**Figure 3 molecules-27-04917-f003:**
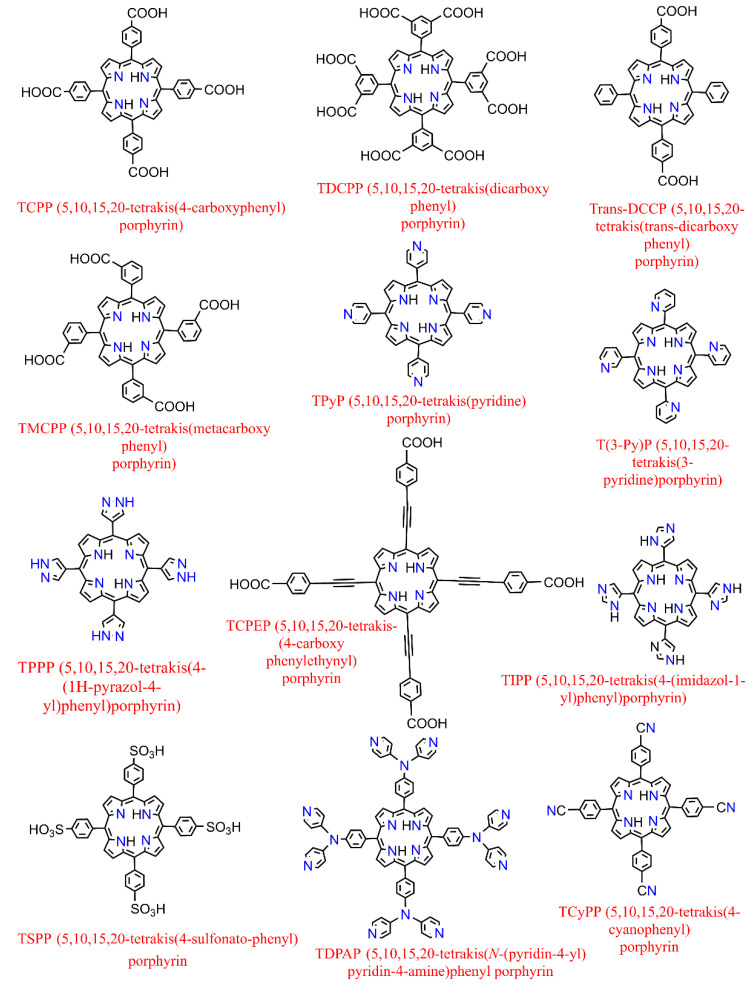
Examples of some of the previously fabricated porphyrin linkers.

**Figure 4 molecules-27-04917-f004:**
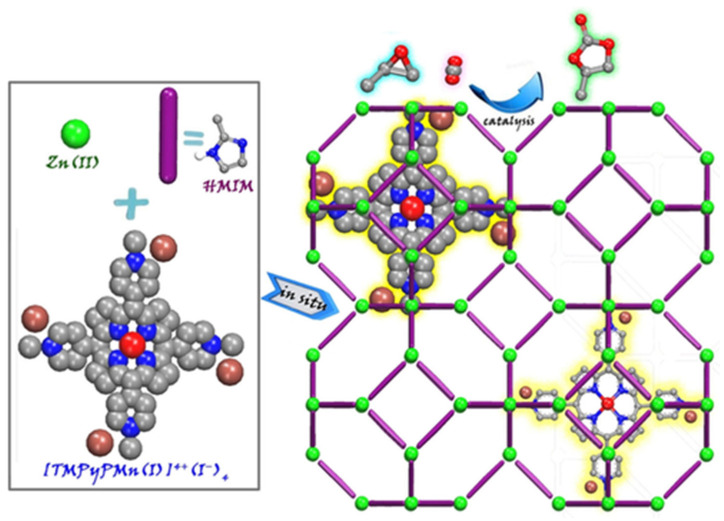
Illustrative demonstration of in situ enveloping of metalloporphyrin into ZIF-8 to conjoin CO_2_ to epoxide. Reprinted with permission from [49].

**Figure 5 molecules-27-04917-f005:**
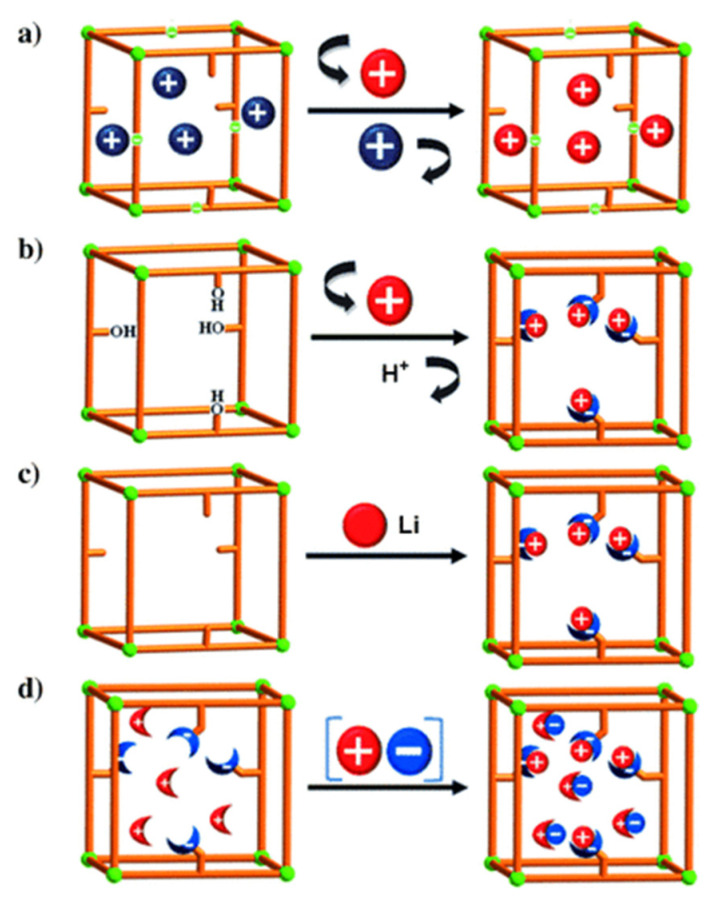
Three basic ways of introduction of open metal sites by PSM synthetic routes to MOFs: (**a**) cationic guests or organic cations exchange (blue balls) with metal cations (red balls); (**b**) replacement of a hydroxy protons with Li^+^ and Mg^2+^ ions (red balls); (**c**) chemical reduction of MOM with Li (red balls) and (**d**) fourth method is a combination of the first two—a collaborative attachment of metal (red balls) chloride (blue ones) salts to anion and cation binding sites. Besides, the sticks and the crescent-shaped bowls attached to sticks are porphyrin-encapsulated inside MOM-11 and cation/anion binding sites. Reprinted with permission from [50].

**Figure 6 molecules-27-04917-f006:**
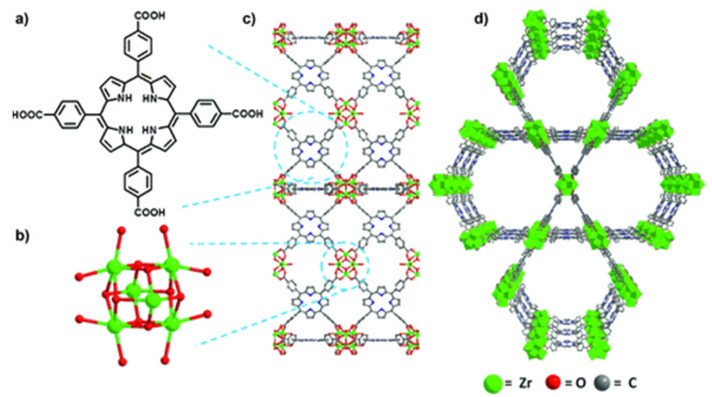
Comparison of free-base PCN-222/MOF-545 (fb-1). (**a**) Tetrakis(4-carboxyphenyl)porphyrin linker, H_4_TCPP. (**b**) [Zr6(m3-O)8(O)8]8¢ node. (**c**) MOF fb-1, shown across the axis a (**d**) 3D structure of fb-1, depicted along the c axis. For more clarity hydrogen atoms has been omitted. Reprinted with permission from [54].

**Figure 7 molecules-27-04917-f007:**
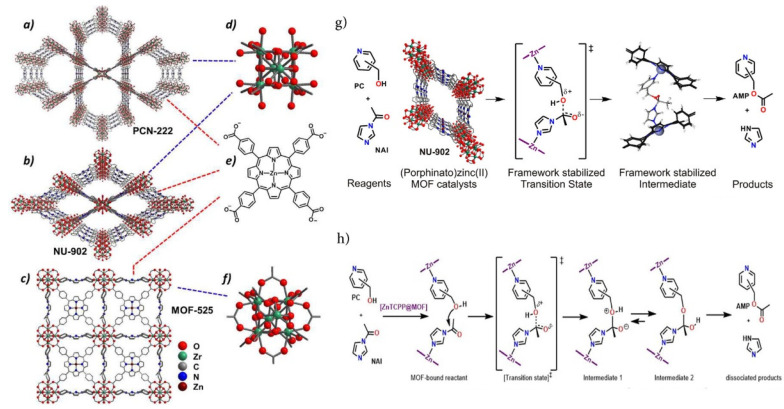
Molecular architecture of (**a**) PCN-222, (**b**) NU-902, and (**c**) MOF-525. (**d**–**f**) Attributed Zr_6_-oxo nodes and the linker (**e**) carboxylate form of Zn − TCPP) are presented on the right, and (**g**,**h**) Lewis acid-catalyzed acyl transfer reaction between pyridylcarbinol (PC) and N-Acylimidazole (NAI) performed by Zirconium-Based (Porphinato)zinc(II) MOF. Reprinted with permission from [80].

**Figure 8 molecules-27-04917-f008:**
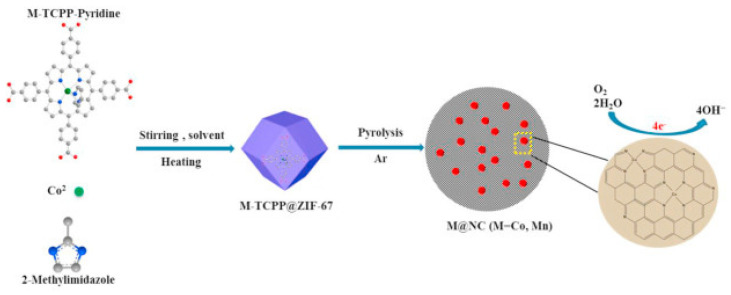
Fabrication of M@NC (M = Mn and Co) catalysts used for electrocatalysis oxygen reduction reaction in alkaline medium. Reprinted with permission from [59].

**Figure 9 molecules-27-04917-f009:**
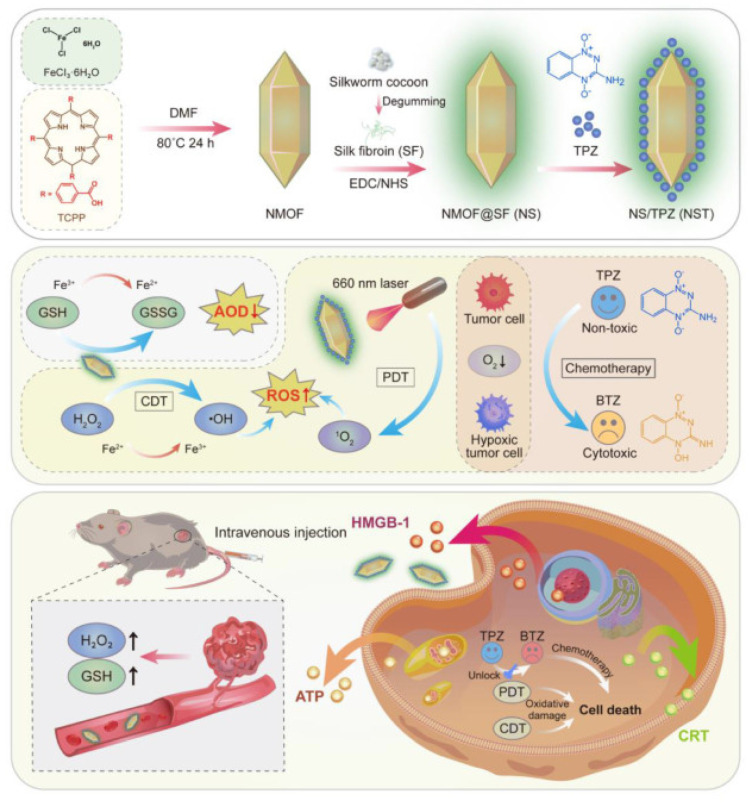
Schematic presentation of the construction procedure of NMOF@SF NPs and their practical mechanism for tumor-specific redox chemodynamic therapy (CDT) combined with photodynamic therapy (PDT) created by Fe (III)-TCPP and glutathione (GSH) upon laser irradiation. Reprinted with permission from [65].

**Table 1 molecules-27-04917-t001:** Naturally occurring metalloporphyrin complexes [27].

Metal Ion	Ionic Radius (ppm) *	Naturally Occurring Complex
Mg^2+^	72	Chlorophyll
Ga^3+^	62	Gallium(III) porphyrin complexes have been found in crude mineral oil but not in living organisms
(V=O)^2+^	≈60	Vanadyl porphyrins are relatively abundant in certain crude oil fractions but they have not been observed in living organisms
Fe^2+^ high spin Fe^2+^ low spin Fe^3+^ high spin Fe^3+^ low spin	78 (too large) 61 65 55 (rather small)	Fe*^n+^* in various oxidation and spin systems is present in heme systems such as hemoglobin
Co^2+^	65	Cobalamins (vitamin B_12_)
Ni^2+^	73	Cofactor F_430_ (catalyzes the reaction that releases methane in the final step of methanogenesis in archaea), tunichlorin

* The ideal ionic radius for a proper *in-plane* coordination is 60–70 ppm. There have also been prepared many artificial complexes containing mostly Mn^3+^ (≈60 ppm), Cu^2+^ (73 ppm) and Zn^2+^ (74 ppm).

**Table 2 molecules-27-04917-t002:** Representative list of metalloporphyrin metal–organic frameworks and their applications.

MOF	Porphyrin	Another Component	Synthetic Procedure	Application	Refs
Pt(II)TMPyP@rho-ZMOF (In)	TMPyP	4,5-H_3_ImDc	In situ	Anion sensing	[40]
M-TCPP@Cu-MOFs (M = Fe, Ni, Co)	TCPP	H_3_BTC	In situ	Electrochemical CO_2_ conversion	[48]
[TMPyPMn(I)]^4+^(I^−^)_4_@ZIF-8	TMPyP		In situ	CO_2_ transformation	[49]
(Mn, Co)-TCCP@ZIF-67	TCPP		In situ	Electrochemical O_2_ reduction	[59]
CoTMPP@ZIF-8	TMPP		In situ	Water oxidation	[60]
PC-MOFs (Zr)	TCPP	Cypate	In situ	PDT/PTT	[61]
Fe(Salen)@PIZA-1 (Co)	TCPP		In situ	OER	[62]
Fe-TPP@ZIF-8-L	TPP		In situ	ORR	[63]
FeTCPP@MOF-SA	TCPP	SA	In situ	DNA sensing	[64]
Fe_3_O_4_@CoTHPP@UiO-66	THPP	Fe_3_O_4_	PSM	Oxidation catalysis	[2]
Fe-TCCP@NU-1000	TCPP		PSM	photochemical CO_2_ reduction	[3]
MTX@PCN-221	TCPP		PSM	Drug delivery	[11]
MA-HfMOF-PFC(PFP)-Ni-Zn	PFP and PFC	EDA-maltotriose	PSM	PDT	[25]
NMOF (Fe)@SF	TCPP	GSH	PSM	CDT/PDT	[65]
UiO-66@porphyrin	TPP-SH		PSM	PDT	[66]
FeTCCP@PCN-333 (Fe)	TCPP		PSM	ORR and HER	[67]
PEG–coated-PCN@PL	TCPP	PEG and PL	PSM	Chemo-sonodynamic therapy	[68]
Hf-NU-1000 (Fe)	TCPP		Porphyrinic	Tandem oxidation catalysis	[19]
PCN-222/MOF-545 (fb-1)	TCPP		Porphyrinic	Mustard gas photooxidation	[54]
PCN-601, PCN-602 (Ni)	TCPP		Porphyrinic	C-H bond halogenation	[69]
USTC-8(In, Cu, Co, Ni, Cd)	TCPP		Porphytinic	H_2_ photochemical production	[70]
PCN-601 (Cu, Co, Fe, Ni)	TPPP		Porphyrinic	Photocatalytic CO_2_ reduction	[71]
PCN-224 (Zr)	TCPP	Vancomycin	Porphyrinic	Antibacterial (PDT)	[72]
[Cd_3_(tipp)(bpdc)_2_]·DMA·9H_2_O	TIPP	H_2_bpdc	Porphyrinic	C-C bond formation	[73]
2D-Zr-MOFs	TCPP		Porphyrinic	Photocatalytic polymerization	[74]
Fe-TBP	TBPP		Porphyrinic	PDT	[75]
ZJU-18, ZJU-19 and ZJU-20	TOCPP		Porphyrinic	Alkylbenzenes oxidation	[76]
FTPF (Cu, Nb, Zn)	TPyP	NbOF_5_	Porphyrinic	CO_2_ fixation	[77]
Cu(TCMOPP) and Ni(TCMOPP)	TCMOPP		Porphyrinic	Alkylbenzenes oxidation	[78]

TMPyP = 5,10,15,20-tetrakis(1-meyhyl-4-pyridinio)porphyrin, THPP = 5,10,15,20-tetrakis(4-hydroxyphenyl)porphyrin, TMPP = 5,10,15,20-tetrakis(4-methoxyphenyl)porphyrin, TPP = 5,10,15,20-tetrakisphenylporphyrin, TCMOPP = 5, 10, 15, 20-tetrakis [4-(carboxymethyleneoxy)phenyl]porphyrin, TBPP = 5,10,15,20-tetrakis(*p*-benzoato)porphyrin, TOCCP = 5,10,15,20-tetrakis(3,5-biscarboxylphenyl)porphyrin, PFP = 5,15-bis(4-carboxylphenyl)-10,20-bis(pentafluorophenyl)porphyrin, PFC = 5,15-bis(4-carboxylphenyl)-10,20-bis(pentafluorophenyl)chlorin, H_2_bpdc = biphenyl-4,4-dicarboxylic acid, DMA = N,N-dimethylacetamide, CDT = chemodynamic therapy. PDT = photodynamic therapy, PTT = photothermal therapy, ORR = oxygen reduction reaction, HER = hydrogen evolution reaction, GSH = glutathione, MTX = Methotrexate, PEG = poly(ethylene glycol), PL = piperlongumine, SA = streptavidin, Salen = bis(salicylaldehyde)ethylenediimin.

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
