# Peer review of "Metalloporphyrin Metal–Organic Frameworks: Eminent Synthetic Strategies and Recent Practical Exploitations"

_molecules, 2022, doi:10.3390/molecules27154917_

Round 1
Reviewer 1 Report
In this review, the authors reported recent development of porphyrin-based metal-organic frameworks (MOFs).
They reported the new synthesis of porphyrin-based MOFs and present-day application of the MOFs such as catalysts and biological utilization by gathering many recent papers.
This review is useful for new researchers to develop the field of porphyrin-based MOFs.
I think that in the any field of science there is the starting point and then the field is developed to present-day prosperity. The process of the development of the field of science is useful for young researchers.
This review would be even better if the starting point of porphyrin-based MOFs is described briefly at “Introduction”.
There are some minor comments as follows.
1. Line 138
MOM-11. ?
2. Fig.3 c)
The figure is obscure.
3. Line 293-299
If the simple figures for three types of porphyrin-based MOFs are added, the three classification is understood more easily.
4. The following paper may offer for reference to staring point of porphyrin-based MOFs.
a) T. Sato, et.al, Microporous rhodium (II) 4,4’,4’’,4’’’- (21H,23H-porphine-5,10,15,20-tetrayl) teteakisbenzoate, synthesis, nitrogen adsorption properties and catalytic performance for hydrogenation of olefin, Chem. Lett. 2003, 32(9) 854-855.
b) T. Sato, et. al, Novel microporous rhodium (II) carboxylate polymer complexes containing metalloporphyrin: syntheses and catalytic performances in hydrogenation of olefins, J. Catalysis, 2005, 232, 186-198.
Author Response
Dear reviewer
Many thanks for your valuable comments that were helpful to enhance our manuscript.
1. We changed and corrected whatever you expressed including all the typos and abbreviations you requested.
- The Figure 5 (in the revised version; in the first version it was Figure 3) has been presented clearly either inside the text or figure caption with their references.
- All the figures related to all kinds of porphyrin-based MOF have been granted a permission and were placed at the correct positions in the revised manuscript (Figures 2, 4, 5, 6, 7, 8 and 9).
- Additionally, we have included the references you have mentioned along with their corresponding descriptions which were incorporated in the manuscript. The new references are [27-37] referring to a new paragraph in lines 69-76, respectively.
Yours sincerely,
Authors
Reviewer 2 Report
This paper summarizes the progress in synthesis and study on the properties of metalloporphyrin-based MOFs. With representative examples, this paper would be quite helpful for researchers interested in the topic. I recommend its publication in Molecules after addressing the following points:
1) The main drawback is that the review is more a set of facts; however it would be good to see some analysis as well.
2) The Introduction section should include background information about the role of metalloporphyrins and MOFs in addressing industrially important conversions. Moreover, the Conclusions section says «These approaches not only improve their stability, but they also result in better performances in (photo-) catalysis, electrocatalysis and mimicking biological systems», however, no specific examples to confirm this statement were given in the main text. It would be interesting to see a comparison of properties, i.e. (photo)catalytic or bio, of MOF, porphyrin and porphyrin@MOF.
3) There are some typos in the text: Nature (line 39) and Ethanol (line 125) should not be written with the capital letters.
4) Section 2.2, which should be devoted to the methods of synthesis, contains a description of the catalytic properties, which should be discussed later in section 3.
5) Some points regarding Table 2:
-As for me, there is no need to give the column «another component»;
- It would be more convenient to combine lines by subgroups (method of synthesis, type of porphyrin, etc.);
- Be more precise in terms of type of catalytic reactions. For example, there is no such a term like «C-C Bond Catalysis», its catalysis of C-C bond formation or, namely, the Knoevenagel condensation and cyanosilylation of aldehydes. In the case of oxidation catalysis, please, indicate the substrate and type of oxidant.
- It would be more convenient if all the types of porphyrins, which are discussed in Table 2, were depicted in Figure 2.
6) Do not re-explain the already introduced abbreviations (for example, TCCP in Line 220).
Author Response
Dear reviewer
Many thanks for your valuable comments that were helpful to enhance our manuscript.
1. With respect to your precious statements, we have decided and made all our efforts to provide a fundamental manuscript explaining the prevalent synthetic routes of metalloporphyrins-based MOFs and some of their recent applications specifically in catalysis, (photo-) electrocatalysis, and biomedical ones where we supposed that we have addressed most of the late ones properly.
- As regards, we addressed and allotted the illustrations and their references in the Introduction part featuring the most relevant topics around metalloporphyrins and MOFs applications even though we did not succeed to find any industrially important conversions to append. Besides, we did not want to emphasize their comparison either with the other materials or their counterparts.
- We have checked the manuscript and made corrections of the typos.
- Regarding the fourth comment, the section has been divided into their relative parts even though we have tried to afford to sort out and report everything appropriately. You could see several literature reviews which they have demonstrated their applications even in the synthesis section, however, we have prepared all of the things in their respective formats and divisions.
- Concerning table 2, the “another component” column should remain because some components should be displayed as they may help the readers to broaden their perspectives about the applications. As we emphasized at various places in the manuscript, the selection of “another components” in the synthesis plays a crucial role in the final properties of a MOF.
We appreciate that you recommend splitting Table 1 into groups and we categorized the data based on the fabrication method in the revised version of the manuscript.
The other applications you mentioned are covered in other papers cited in the Introduction section. Indeed, it is not possible to mention every application and we have chosen the most important ones.
For the porphyrins linker that we put in the table, we wanted to introduce some porphyrins ligands in the Table with their utilizations and some of the ordinary previously-synthesized ones in Figure 3 (in the revised version) in a different way to give the reader a more extensive insight into the whole issue.
- We erased all the extra abbreviations that we have elucidated beforehand in the context.
Yours sincerely,
Authors
Reviewer 3 Report
Krivosudský and coauthors describe the pertinent synthetic processes by which porphyrin-based MOFs are fabricated and some of their popular new-fashioned utilization. The paper is well written, while the flow of the paper needs to be revised and some important references and examples need to be added. Section 3 needs to be expanded as the current one is not very detailed. Therefore, the following question must be addressed, before I can reconsider it for publication.
1. For section 3, it is fine that the authors listed the examples in the Table. However, a better way is to use figures to show some of the important examples. Especially, for the examples that are discussed in the main text, some figures/schemes could better help readers understand the paper. I strongly suggest the authors take the important figures from those papers, get the copyright, and combine the related examples in a couple of figures.
2. Section 3.1 and 3.2 are both talking about catalysis application, while section 3.3 is talking about biomedical application, and I ain’t quite understand the logic here. My understanding is the subtitle of section 3 should be for different applications, while 3.1 and 3.2 are talking about one application. Please either make changes or give the reasons.
3. Following question 2, is there any other application that can be introduced? As far as I know, there are a lot of applications that metal porphyrin-based MOFs could do. For example, ion absorption (water treatment), battery and CO2 capture. I do not think the authors should include all the examples, but those applications should be at least introduced briefly.
4. Considering the similarity of COF and MOF, some of the recent examples of porphyrin-based COF should be also briefly introduced and cited. (https://doi.org/10.1039/D1SC05379E; https://www.nature.com/articles/s41467-021-21527-3). These could be either in the introduction or outlook or as the example in Section 3. This way a bigger picture could be presented to the readers.
5. Some sentences are missing references. Please check whether there are more.
a. “post-synthetic method containing anchoring to the exterior or inclusion of them inside the MOF pores are mainly based on weak chemical interactions such as hydrogen bonding, electrostatic, van der Waals forces and others, between the pre-obtained MOF and porphyrin base/metalloporphyrin.”
b. “they have chiefly concluded that resolving would be possible through immobilization of porphyrin inside or anchoring them to the surface, and as struts in MOF framework.”
c. “catalytic performances lying completely in MOF architecture in comparison with 86 their homogeneous counterparts”
d. “As a consequence, the catalytic performances of these types of porous coordination networks could be simply upgraded by regulating Lewis-acid metal active sites and designing well-qualified circumferential functionalities on metalloporphyrins as well.”
6. Is that possible to use figures or schemes to show the difference between in situ or grafting on the surface method to make Porph@MOFs? Just using text might be hard to express and understand.
7. Some typos in the manuscript and the authors should double-check. For example, “Ethanol” in line 125, “E” should be “e”.
8. Figure 3 is quite confusing, and the drawing is rough. Some text should be added to better express what the authors want to say. For example, what is the difference between pink “+” and yellow “+” in Figure 3a. Besides, M+ should be used to show metal cation instead of just “+”, as “+ “ could also refer to positive charge, hole and others.
9. Figure 3c, what are the red balls and sticks represent? Be more specific.
10. Figure 3c, how does the Li create the -o- on the MOF? On which part of MOF? And how is this related to porphyrin grafting?
11. The text for Figure 3 should be revised. Figure 3a 3b and 3c should be all discussed and referred to separately. Currently, the authors just say “ Furthermore, submerging single crystals of the prefabricated MOFs into metal chloride salt solutions, with meso-tetra(N-methyl-4-pyridyl) porphine tetratosylate (TMPyP) in methanol serving as the template for PSM, formed a basis for a MOF formation via single-crystal-to- single-crystal ion exchange processes (Fig. 3)” and refer to the whole Figure 3, which is not specific enough and confusing. Each synthesis route must be discussed in detail.
Author Response
Dear reviewer
Many thanks for your valuable comments that were helpful to enhance our manuscript.
1. We got permissions for some new figures – Figure 2, 4, 5, 6, 7, 8, and 9. They have been placed in appropriate paragraphs throughout the manuscript and designated with explanatory figure captions.
- Section 3.1 and 3.2 are not the same – Section 3.1 former is devoted to catalysis and Section 3.2 deals with (photo-) electrocatalysis. Hence, the MOFs discussed as examples are entirely different.
- Clearly, there are many other applications apart from those introduced in the manuscript and those suggested in the reviewer´s comment. We wished to discuss the most important ones and the other applications are discussed in the individual papers mentioned in the Introduction section.
- We here have only concentrated to provide metalloporphyrins-based MOF, and not related COF. This would be a topic for an individual review.
- For clarification, all the references were placed at the end of their determined sentences as suggested.
- As described above, we situated all the schemes/figures and the publishers granted us permission to use the figures in this manuscript.
- We have checked out that all the typos were removed as well as other errors.
8-11. For Figure 5 (in the revised manuscript) we have obliterated and added some pertinent descriptions either at bottom of the figure or the ones in which we discussed each section separately and also referred to them as represented in lines 162-184. All the comments necessary for clarification were included.
Yours sincerely,
Authors
Round 2
Reviewer 3 Report
Not all questions are well addressed but current version is still ok to publish